# Design and Implementation of a pH Sensor for Micro Solution Based on Nanostructured Ion-Sensitive Field-Effect Transistor

**DOI:** 10.3390/s20236921

**Published:** 2020-12-03

**Authors:** Yiqing Wang, Min Yang, Chuanjian Wu

**Affiliations:** State Key Laboratory of Materials-Oriented Chemical Engineering, College of Electrical Engineering and Control Science, Nanjing Tech University, Nanjing 211816, China; yangmin@njtech.edu.cn (M.Y.); wuchuanjian@njtech.edu.cn (C.W.)

**Keywords:** ion-sensitive field-effect transistor, nanostructure, pH sensor, performance analysis

## Abstract

pH sensors based on a nanostructured ion-sensitive field-effect transistor have characteristics such as fast response, high sensitivity and miniaturization, and they have been widely used in biomedicine, food detection and disease monitoring. However, their performance is affected by many factors, such as gate dielectric material, channel material and channel thickness. In order to obtain a pH sensor with high sensitivity and fast response, it is necessary to determine the appropriate equipment parameters, which have high processing cost and long production time. In this study, a nanostructured ion-sensitive field-effect transistor was developed based on the SILVACO technology computer-aided design (TCAD) simulator. Through experiments, we analyzed the effects of the gate dielectric material, channel material and channel thickness on the electrical characteristics of the nanostructured field-effect transistor. Based on simulation results, silicon nitride was selected as the gate dielectric layer, while indium oxide was chosen as the channel layer. The structure and parameters of the dual channel ion-sensitive field-effect transistor were determined and discussed in detail. Finally, according to the simulation results, a pH sensor based on the nanostructured ion-sensitive field-effect transistor was fabricated. The accuracy of simulation results was verified by measuring the output, transfer and pH characteristics of the device. The fabricated pH sensor had a subthreshold swing as low as 143.19 mV/dec and obtained an actual sensitivity of 88.125 mV/pH. In addition, we also tested the oxidation reaction of hydrogen peroxide catalyzed by horseradish peroxidase, and the sensitivity was up to 144.26 pA mol^−1^ L^−1^, verifying that the ion-sensitive field-effect transistor (ISFET) can be used to detect the pH of micro solution, and then combine the enzyme-linked assay to detect the concentration of protein, DNA, biochemical substances, biomarkers, etc.

## 1. Introduction

Biosensors based on ion-sensitive field-effect transistors (ISFETs) have wide potential applications and are expected to be the preferred devices for field diagnosis. The ISFET was first proposed by Piet Bergveld in 1970 and used for pH detection [1]. It has advantages of fast responses, high sensitivity and simple miniaturization [2]. The Honeywell Durafet is a pH-sensitive Ion-Selective Field Effect Transistor that has been successfully used for pH measurement in seawater CO_2_ chemistry and ocean acidification studies [3,4]. Modified Honeywell ISFETs have also been integrated into a novel solid-state sensor capable of rapid, simultaneous measurements of pH and total alkalinity in seawater [5,6]. In addition, a one-step immunoassay for the detection of influenza A virus was developed by using two different microbead types and a filter-inserted bottle [7]. The detection bead could pass through the filter into the reaction buffer containing urea and induce a pH change, which was quantitatively measured by using phenol red and ISFET. While the one-step immunoassay could be applied to both colorimetry and ISFET assays, the detection range was wider with the ISFET-based one-step immunoassay (0–100 ng/mL). Therefore, pH detection based on ISFETs is very important in the fields of chemistry, agriculture, food processing, pharmacology, environmental science and biomedical engineering [8,9,10,11]. Given that high-performance pH sensors require extremely sensitive and stable operation, dual-gate (DG) ISFET-based pH sensors have been developed to improve their sensitivity and stability. The DG ISFETs can significantly amplify their sensitivity and stability without additional amplification circuits by inducing capacitive coupling effect between the top and back gates of the channel. With the proper combination of top gate capacitance and back gate capacitance, an ISFET with a DG structure can even exceed the Nernst limit (59 mV/pH) [12]. An amorphous-InGaZnO (a-IGZO) thin-film transistor was designed to create a biosensor that can operate with DG mode, with a pH sensitivity of 286.76 mV/pH [13]. An individually addressable back gate and a gate oxide layer directly exposed to the solution were created by using SOI (Silicon-On-Insulator) technology. When the ISFET operated in a DG mode, the response can exhibit sensitivity to pH changes beyond the Nernst limit, which was up to 107.65 mV/pH [14]. A DG silicon nanowire (SiNW) FETs pH sensor was demonstrated that can be operated in amplification mode by employing an inversed DG configuration that used the electrode immersed in a solution as the top gate and the SiNW metal gate as the back gate [15]. By integrating the sensor chip with complementary metal oxide semiconductor (CMOS) circuits, the sensitivity of dual-amplification mode obviously exceeded the Nernst limit. The DG ISFET is significantly improved in terms of sensitivity, signal-to-noise ratio, drift rate and hysteresis. This will play a strong role in promoting medical testing, environmental protection and industrial production [16,17,18,19,20,21,22].

In recent years, with the development of material science and photolithography technology, ISFETs have gradually become more miniaturized and integrated and can be used for pH detection of trace solutions [23]. Ion-sensitive metal oxides (MO_X_) with nanostructures have excellent biocompatibility. ISFET devices based on MO_x_ are often applied to the detection of various ion concentrations. Thus, they are widely employed in the fields of electrochemical sensing, biomedical science, water and food quality monitoring, disease detection and other fields [24,25,26]. Among them, pH sensors used to detect the concentration of hydrogen ion (H^+^) are the most widely used [27,28]. An ISFET-based pH sensor utilizing a low-cost polyethylene terephthalate (PET) film coated with Au/Ta_2_O_5_ layers as the extended gate electrode was developed [29]. The extended gate structure was connected to the gate of a commercial metal–oxide–semiconductor field-effect transistor (MOSFET) integration circuit (IC) for H^+^ concentration measurement with the sensitivity of 24.18 mV/pH in the pH range from 1 to 13. Moreover, an ISFET biosensor based on enzyme modification was proposed [30]. Utilizing the newly prepared CMOS-compatible ISFET with Ta_2_O_5_ sensitive surface and adopting the subthreshold working mode, high sensitivity measurement was realized by reducing the influence of capacitance on the subthreshold swing value. The pH sensitivity of the prepared ISFET was 52 mV/pH. In addition, a high-sensitivity normally off MOS-ISFET pH sensor was developed by pulse electrochemistry (PEC) method. The AlGaN barrier in the sensing region was transformed into an oxide layer with a thickness of about 20 nm to achieve normally off operation. As a result, its pH sensitivity was increased to about 56.3 mV/pH [31]. It can be seen that pH sensors can improve sensitivity by improving parameters and manufacturing methods. However, pH sensors based on semiconductor field-effect transistor (FET) have high processing cost and long production time. Moreover, many researchers choose to optimize performance through simulation. An ISFET-based pH sensor model with temperature-dependent behavioral macromodel was presented [32]. The macromodel was built by using simulation program with integrated circuit emphasis (SPICE), by introducing electrochemical parameters in a MOSFET model to simulate ISFET characteristics. The framework for integration of machine learning (ML) techniques for drift compensation of ISFET chemical sensor was established to improve its performance. Moreover, a numerical simulation approach to study the electrolyte pH change of ISFET structures, using Silvaco technology computer-aided design (TCAD) tools, was proposed [33]. Additionally, transfer characteristics of a conventional ISFET device were simulated, and the I_D_ current as a function of the reference voltage, V_Ref_., and drain voltage, V_D_, for different pH scale and I_D_ current as a function of V_DS_ for different V_Ref_. values for specific pH value were also simulated. The proposed models allow accurate and efficient ISFET modeling by trying different designs and further optimization with different tools rather than expensive fabrication. However, they did not fabricate specific devices to verify the accuracy of the simulation results. Therefore, in order to shorten the production cycle and obtain a pH sensor with high sensitivity, fast response, low cost and suited for detect micro solution, we decided to use simulation software to design devices, analyze the influence of structure and size parameters on device performance, and optimize device structure and parameters before processing.

In this work, we first designed a double gate FET device based on MO_X_. Then, we used SILVACO TCAD simulation software to analyze and to compare the electrical characteristics of FET devices with different sensitive film materials, oxide nanomaterials and channel size. We further optimized the structure and size of the FET device designed. The feasibility of the simulation ISFET structure was verified by comparing the electrical characteristics of the fabricated ISFET device and the simulation device. Furthermore, commercial pH solution was used to test the sensitivity of the prepared ISFET. Finally, it was used to detect pH changes of the solution caused by the hydrogen peroxide oxidation reaction with horseradish peroxidase to verify its effectiveness and sensitivity.

## 2. Structure of Device and Simulation

The nanochannel FET was simulated by SILVACO TCAD. The cross-section of the In_2_O_3_ nanobelt FET is shown in Figure 1a. The parameters of the nanobelt FET are shown in Table 1. Figure 1b shows the 3D model of the In_2_O_3_ nanobelt device prepared through the simulation. Si was used as the substrate, the insulating layer was Si_3_N_4_, the channel material was composed of the In_2_O_3_ nanobelt, thickness was 50 nm, w/L ratio was 9, and Cr/Au was used as the source–drain electrode material.

## 3. Theoretical Analysis

### 3.1. Threshold Voltage

Under normal conditions, the electric field generated by a gate voltage controls the generation of carriers in the channel region. The threshold voltage ($V_{T}$) of the device is defined as the gate–source voltage when the semiconductor surface at the source end of the channel begins to be strongly inverted. At this time, it will experience a state where the surface electron concentration is equal to the hole concentration and the device is in a critical conduction state. According to the relationship between the gate–source voltage and the surface potential, the $V_{T}$ can be expressed as follows:(1)$$V_{T}=\phi_{CS}+\phi_{S}-\frac{Q_{Ox}}{C_{ox}}-\frac{Q_{B}}{C_{ox}}\;,$$
where $\phi_{CS}$ is the work function potential difference between the chemical film and semiconductor. $\phi_{S}$ is the surface potential of semiconductor, while $Q_{Ox}$ is the effective charge surface density of the oxide layer. $C_{ox}$ is the gate oxide capacitance per unit area and $Q_{B}$ is the charge surface density of the depletion layer.

### 3.2. Source–Drain Current

The current Equation of the FET shows the relationship between drain current and gate voltage and drain voltage. The conduction state of the device is described as follows. The voltage $V_{DS}$ is set to a very small value, and the electric field intensity in the channel is assumed to be much lower than the velocity saturation electric field. The source–drain current, $I_{DS}$, can be expressed as follows:(2)$$I_{DS}=\frac{\mu_{n}C_{ox}W}{L}\left[\left(V_{GS}-V_{T}\right)V_{DS}-\frac{1}{2}V_{DS}^{2}\right]\;(\text{non-saturated}\;\mathrm{region}),$$
(3)$$I_{DS}=\frac{\mu_{n}C_{ox}W}{2L}\left[{\left(V_{GS}-V_{T}\right)}^{2}\right]\;(\mathrm{saturated}\;\mathrm{region}),$$
where $\mu_{n}$ is the carrier mobility while $C_{ox}$ is the gate oxide capacitance per unit area; $V_{GS}$ represents the gate–source voltage and $V_{DS}$ is the source–drain voltage; *w*/*L* stands for the channel width length ratio; $V_{T}$ is the threshold voltage.

In the unsaturated region, when $V_{DS}$ is very small, Equation (2) can be simplified as follows:(4)$$I_{DS}=\frac{\mu_{n}C_{ox}W}{L}\left(V_{GS}-V_{T}\right)V_{DS},$$

At this time, a relatively uniform conductive channel is formed from the source to the drain, and the integral channel is equivalent to a resistance value, which is proportional to a resistance of ($V_{GS}–V_{T})$^−1^. A very large value of the gate–source voltage, $V_{GS}$, can result in small resistance. The current passing through the channel increases linearly with $V_{DS}$, so the characteristic curve in the work area becomes linear. In the linear region, the transconductance of the device is defined as follows:(5)$$g_{m}={\frac{\partial I_{DS}}{\partial V_{GS}}|}_{V_{DS}}=\frac{W}{L}\mu_{n}C_{ox}V_{DS},$$

As $V_{DS}$ increases to $V_{Dsat}=V_{GS}-V_{T}$, the inversion charge *Q_i_* = 0 of drain end reaches to 0. This means that the drain end cannot form an inversion layer, and the channel is clamped at the drain end. With the increase in $V_{DS}$, the pinch point gradually moves from drain to source, and the increased voltage falls in the pinch off area between the pinch point and drain, forming a strong electric field region. After the channel is clamped, there is still a conductive channel between the source and the pinch point, because the voltage in the channel is basically maintained at $V_{Dsat}.$ When $V_{Dsat}$ is constant, the drain current remains unchanged and reaches the saturation value $I_{Dsat}$, where we get the following:(6)$$I_{Dsat}=\frac{\mu_{n}C_{ox}W}{2L}\left[{\left(V_{GS}-V_{T}\right)}^{2}\right],$$

In the saturated region, the following relation exists:(7)$$g_{m}=\frac{W}{L}\mu_{n}C_{ox}\left(V_{DS}-V_{T}\right),$$

In ideal conditions, the transconductance in the saturated region is 0. However, due to the shortening of the effective channel length, there is a non-zero channel conductance in the saturated region.

### 3.3. Subthreshold Swing

The subthreshold swing (*s*) is an important parameter for switching operations. The gate voltage required to increase the drain current by an order of magnitude is defined. The swing reflects the transition steepness of the current from the off state to the on state. This corresponds specifically to the reciprocal of the slope of the subthreshold line segment in the device transfer characteristic curve using semilogarithmic coordinates, which can be expressed as follows:(8)$$S=\frac{dV_{GS}}{d\left(\log I_{DS}\right)},$$

In current development cycles, with intent to develop point-of-care test (POCT) equipment [34], whose power supply module is only about 5 V, the characteristic size of the device is constantly shrinking. It requires the device to work under low voltage and lower the threshold voltage, which is convenient for us to use ISFET for portable detection. In addition, in order to ensure a certain response speed, it is necessary to produce a steep subthreshold slope to reduce the off-state current.

## 4. Results and Discussion

The ATLAS device simulation tool with different parameters was used to study the performance of the proposed In_2_O_3_ nanobelt ISFET, and the optimal device structure and parameters were thereby determined. The models used to the device simulation include the concentration dependent mobility model, Shockley–Read–Hall (SRH) composite model and Fermi Dirac statistical model.

### 4.1. Description of Electrical Characteristics of Devices

First, the output characteristics of the device were tested. As shown in Figure 2a, the applied gate voltage was varied. The source–drain current varied with increase in the source–drain voltage. The gate voltage value ranged from 0.5 to 1 V with a step size of 0.1 V. The source–drain current changes as the source–drain voltage was varied from 0 to 2 V. It is revealed that large applied gate voltages can result in large corresponding source–drain currents. This follows the drain current analyzed in the previous section. The source–drain current can be enhanced with increase in gate voltage, after the source–drain voltage reaches a certain value. The source–drain current can saturate, which is consistent with the theory. When $V_{GS}=1.0\;V$, the source–drain current can reach 100 μA. Figure 2b shows the transfer characteristics and transconductance of the device, which clearly indicates the gate voltage’s ability to regulate source and drain current. Setting the source–drain voltage to 2.5 V, and scanning the gate voltage from −1 to 1 V, we can obtain the transfer characteristic curve, and draw the transconductance curve according to the transconductance Equation (8) analyzed in the previous section. Through the built-in data extraction function in SILVACO TCAD, the threshold voltage can be easily obtained as −0.24 V as $V_{DS}$ = 2.5 V. In addition, according to the transfer characteristic curve, the device-switching current ratio can reach as high as 10^8^. Figure 2c shows the semilogarithmic curve of the relationship between the source and drain current of the device under the condition of $V_{DS}$ = 2.5 V. According to the theoretical analysis in the previous section, when the drain current rose by an order of magnitude, that is, from 0.01 to 0.1 pA, the gate–source voltage changed from −0.95 to −0.87 V. It can be concluded in terms on Equation (8) that the subthreshold swing was 81.38 mV/dec, which was slightly higher than theoretical value. This value was a result obtained under an ideal simulation environment. Realistic devices may not achieve such good performance. Actually, the subthreshold swing of the device fabricated in this work was 143.19 mV/dec, which would be described in Section 4.4.

### 4.2. Material Performance Analysis

#### 4.2.1. Material Analysis of a Gate Dielectric Layer

Figure 3a shows different gate dielectric materials at *V**_GS_* = 0.5 V. From the results, as the source–drain voltage was scanned from 0 to 2 V, the gate dielectric material Si_3_N_4_ had a more defined linear region and saturation current than SiO_2_. This is because the dielectric constant of SiO_2_ is 3.9, while the dielectric constant of Si_3_N_4_ can reach 7.5 [35,36]. Furthermore, its bandgap width is lower than that of SiO_2_. Therefore, the large capacitance value per unit area of Si_3_N_4_ can result in a large corresponding source–drain current. Figure 3b shows the application of the same source–drain voltage ($V_{DS}=2.5\;V$). The semilogarithmic curves of drain current and gate voltage of different gate dielectric materials can be calculated using the theory in the previous section. When the gate dielectric material was SiO_2_, the subthreshold swing of the device was 100.60 mV/dec, which was larger than that when the gate dielectric material was Si_3_N_4_, leading to slow work rate in the subthreshold region.

As a sensitive membrane material, SiO_2_ has some inherent disadvantages. It is subject to diffusion of protons or hydrogen, and source–drain current, resulting in serious drift and deterioration of ISFET based on SiO_2_ [37,38,39]. Compared with other high-k sensitive film materials, Si_3_N_4_ has many advantages. For instance, it is free from interference impurities, the thickness of the film can be controlled, and no transition layer is required because of favorable conditions in the interface between Si_3_N_4_ and substrate Si [40,41,42]. Therefore, Si_3_N_4_ is preferred as the sensitive layer to obtain better device performance.

#### 4.2.2. Analysis of Semiconductor Material

Zinc oxide (ZnO) and indium oxide (In_2_O_3_) are widely used as channel materials for electronic devices due to their chemical stability and high carrier mobility [43,44]. As a natural self-doped n-type semiconductor, it is easy to crystallize, which is conducive to obtain high carrier mobility films, and can be deposited by various coating methods. The difference in process technology also affect device performance. Taking this into consideration, we used Athena to form the structure under the same process technology and conditions, and then compared the electrical characteristics by Atlas. Figure 4a,b, respectively, shows the output and transfer characteristics of the device when the channel materials were ZnO and In_2_O_3_. The output characteristic curve was obtained by setting the gate voltage to 0.5 V and scanning the source–drain voltage from 0 to 2 V. The transfer characteristic curve was obtained by setting the source–drain voltage at 2.5 V and scanning the gate voltage from −1 to 0 V. It can be concluded that compared with ZnO, the In_2_O_3_ film can obtain a clearer source–drain current, smaller off-state current and has better performance. The on/off current ratio of In_2_O_3_ was 10^8^ while ZnO was 10^5^. In_2_O_3_ also has lower process temperature and higher field mobility [45,46,47], so we used In_2_O_3_ as the channel material in device fabrication.

### 4.3. Influence of Channel Parameters on Device Performance

Through comparison of device performance between ZnO and In_2_O_3_, In_2_O_3_ was finally selected as the channel material. In order to explore the influence of the thickness and length of the In_2_O_3_ channel layer on the performance of the FET, a range of values were simulated. First, five thicknesses, namely 0.02, 0.05, 0.1, 0.12 and 0.15 μm, were selected to compare the output characteristics and transfer characteristics of the device. When testing the output characteristics, the gate voltage was set to 0.5 V, while the source–drain current was scanned from 0 to 2 V. When testing the transfer characteristics, the source–drain current was set to 2.5 V, and the gate voltage was scanned from −1 to 0 V. The simulation results are illustrated in Figure 5a,b. Moreover, the comparison of subthreshold swing at different thicknesses are shown in Table 2. From the Figure 5a, it is clear that the source–drain current increased with an increasing thickness of the In_2_O_3_ channel. As shown in Figure 5b, the turn-off current increased significantly when the thickness reached more than 0.05 μm. If the channel thickness is too large, carriers still flow under the action of the source–drain voltage when the device is turned off which resulting in a larger leakage current. Moreover, as can be seen from Table 2, the larger the thickness of the In_2_O_3_ channel, the greater the device subthreshold swing and the smaller the turn-on and turn-off rate. Therefore, considering device performance and manufacturing process in combination, the thickness of the In_2_O_3_ channel layer was defined as 0.05 μm. In addition, when the channel width is constant, the difference in channel length will also cause changes to device performance. As shown in Figure 6a, the output characteristics of different channel lengths 0.4 and 0.8 μm were measured when the gate voltage was set to 0.5 V. It is clear that when the channel width is constant, a short channel length results in large source–drain current. This also satisfies the relationship between the source–drain current and the aspect ratio analyzed in the previous section, i.e., the larger the aspect ratio, the greater the channel length. Fast carrier movement rate results in large source–drain current. Figure 6b shows the transfer characteristics of different channel lengths when the source–drain voltage was 2.5 V. The threshold voltages of the two were similar. The large channel-width-to-length ratio resulted in a large switching current ratio, indicating strong capability of the device to regulate current.

### 4.4. Device Performance Verification

According to abovementioned simulation and analysis results, the In_2_O_3_ nanobelt ISFET based on the proposed structure was fabricated by using the processing flow shown in Figure 7. First, Si_3_N_4_ film was grown on Si substrate by radio frequency sputtering. Then, a photoresist was coated on the Si_3_N_4_ film for forming source and drain masks after photolithography. Next, metal was deposited by electron beam evaporation. After peeling off the photoresist, we formed the source and drain. Another photoresist was coated continuously, to create the nanobelt mask. Finally, In_2_O_3_ was deposited by magnetron sputtering and the nanobelt was formed, resulting in fabrication of nanostructured field-effect transistors. The structure of the ISFET sensing region is illustrated in Figure 8. The bandwidth ratio of In_2_O_3_ is 9 nm, and its thickness is 50 nm.

As shown in Figure 9a,b, the output and transfer characteristics of the device were detected for the prepared In_2_O_3_ nanobelt ISFET through a semiconductor parametric test system (PDA, FS380), which was consistent with the simulation results. When the source–drain voltage rose to a certain value, the source–drain current reached saturation. According to the transfer characteristic curve, the subthreshold swing of the actual device can be calculated by Equation (8) as 143.19 mV/dec. In addition, standard commercial pH (pH 6–10) solutions were used to measure the transfer characteristics. The detection equipment is shown in Figure 10a, where the reaction cell was 2 mL. The ISFET was fixed under the container. The Ag/AgCl reference electrode was placed in the solution as the top gate. Figure 10b shows the DG operating method. Biased voltage V_GS_ = 4 V was applied to the bottom Si substrate as the back gate while the top Ag/AgCl reference electrode was grounded. The voltage between the drain and source V_DS_ was set to 1 V. We added a 1 mL solution with different pH values respectively. The measurement results are shown in Figure 11a. From the results, the source–drain current gradually increased with increase in pH value. Taking the gate–source voltage corresponding to the source–drain current of 8 μA, as shown in Figure 11b, there was a linear relationship between the gate–source voltage and pH from 6 to 10, and the regression Equation was V (voltage) = −0.088125 pH + 1.55375 (R^2^ = 0.98635). The calculated sensitivity of the In_2_O_3_ nanobelt ISFET can reach 88.125 mV/pH. In order to further describe the sensitivity performance of the pH sensor, take the relationship between the drain–source current and pH when the gate–source voltage was 1.2 V. As shown in Figure 11c, the drain–source current had a linear relationship with pH between 6–10. The regression equation was I (current) = 6.55 pH − 18.42 (R^2^ = 0.99534) and the sensitivity was 6.55 μA/ pH. It can be seen that the prepared In_2_O_3_ nanobelt ISFET can detect the change of pH value of trace solution.

When the enzyme substrate undergoes oxidation or hydrolysis under the action of the enzyme, the OH^−^ concentration in the solution will change, which will cause a change in pH. Therefore, in order to verify that the device can detect the pH change of micro solution furtherly, we used the hydrogen peroxide oxidation reaction under the catalyze of horseradish peroxidase to change the pH value of micro solution. The reaction proceeded as follows:(9)$$H_{2}O_{2}\overset{\mathrm{HRP}}{\to }H_{2}O+{\mathrm{OH}}^{-},$$

The 30 U horseradish peroxidase was immobilized on the surface of ISFET and placed at 4 °C for 24 h. Different amounts of hydrogen peroxide (100 μL, 200, 300, 400 and 500 μL) at a concentration of 0.1 mol L^−1^ was added in 0.01 × phosphate buffer saline (PBS), the total volume was 1 mL. When the hydrogen peroxide oxidation reaction proceeded, OH^−^ ions were released into the solution, as shown in Equation (9), and the pH value of the solution increased. As the hydrogen peroxide content increased, the drain–source current gradually increased. The curve is shown in Figure 12a. As shown in Figure 12b, the source leakage current gradually increased with the increase of the hydrogen peroxide content. There was a linear relationship between the source leakage current and the hydrogen peroxide content from 100 to 500 μL, and the regression equation was I (current) = 0.144265C (hydrogen peroxide) + 12.83788 (R^2^ = 0.99738). The calculated sensitivity of urea detection was 144.26 pA mol^−1^ L^−1^. The above experiment proved that the ISFET developed in this work can detect the pH change caused by the oxidation reaction of enzyme substrate in the trace solution. Based on this conclusion, labeling the antigen or antibody with enzyme and adding the corresponding enzyme substrate, the prepared ISFET can be used for enzyme-linked reaction to detect the concentration of protein, DNA, biochemical substances, biomarkers, etc. In conclusion, the simulated device structure model used to develop the device has significant potential in the application of pH sensors.

## 5. Conclusions

In this paper, the two-dimensional simulation of a nanochannel FET was studied by using the SILVACO TCAD simulation software. By analyzing the threshold voltage, source–drain current and subthreshold swing model, the influences of gate dielectric material, channel material and channel size on device performance were systematically considered. For final fabricated device, Si_3_N_4_ was selected as the sensitive film material, In_2_O_3_ was used as channel material, the channel thickness was determined to be 50 nm and the width-to-length ratio was 9. By comparing the measured device characteristics with the simulated data, the accuracy of the simulation structure was verified. Moreover, the structure can be used for pH detection with a sensitivity of 88.125 mV/pH and hydrogen peroxide oxidation reaction with a sensitivity of 144.26 pA mol^−1^ L^−1^. In conclusion, the proposed structure can be used to detect biomolecules based on pH detection and has promising applications for water and food quality control, chronic disease treatment and industrial production.

## Figures and Tables

**Figure 1 sensors-20-06921-f001:**
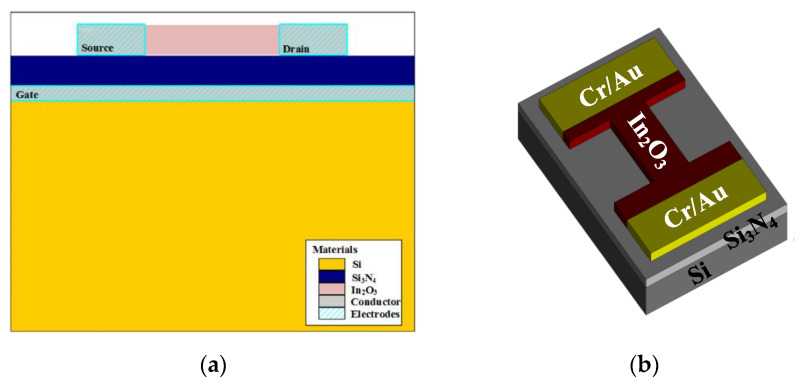
In_2_O_3_ nanobelt field-effect transistor (FET): (**a**) FET cross-section of In_2_O_3_ nanobelt and (**b**) FET 3D model of In_2_O_3_ nanobelt.

**Figure 2 sensors-20-06921-f002:**
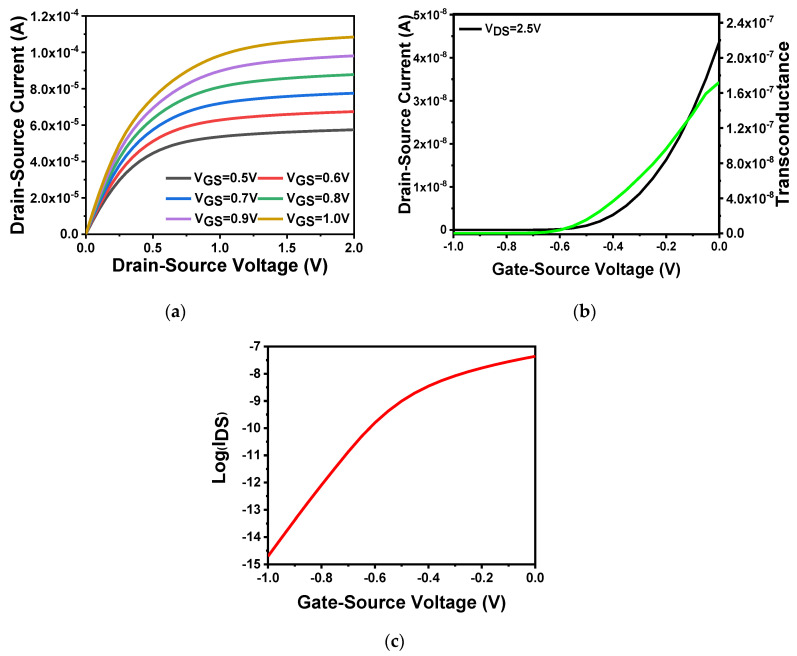
In_2_O_3_ nanobelt FET characteristics: (**a**) device output characteristic curve; (**b**) the black curve is the transfer characteristic curve corresponding to the left coordinate, and the green curve is the transconductance curve corresponding to the right coordinate; (**c**) semilogarithmic curve of the relationship between source–drain current and gate–voltage (*V_D_**_S_* = 2.5 V).

**Figure 3 sensors-20-06921-f003:**
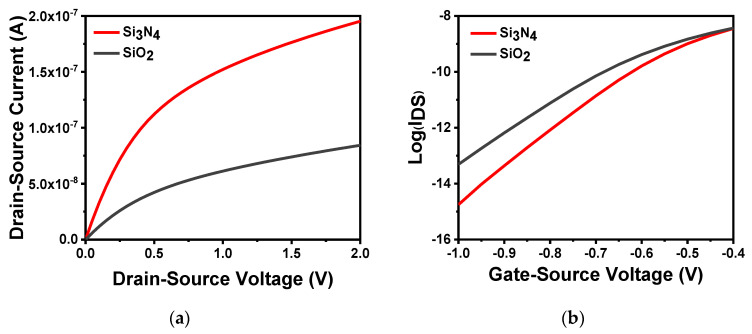
Characteristics of different gate dielectric materials: (**a**) output characteristic curve of different gate dielectric materials in *V**_GS_* = 0.5 V; (**b**) different gate dielectric materials semilogarithmic curve of drain current versus gate voltage at *V_D_**_S_* = 2.5 V.

**Figure 4 sensors-20-06921-f004:**
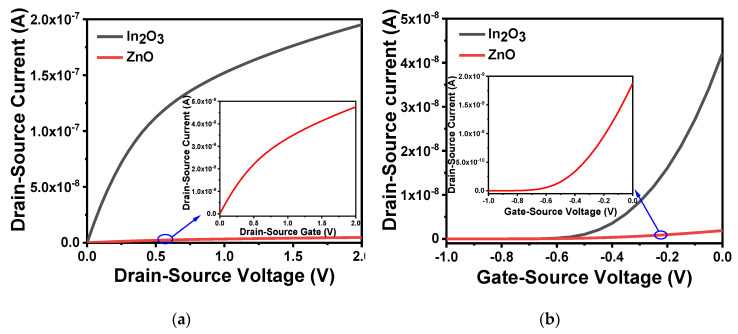
Characteristics of different channel materials: (**a**) output characteristic curve of different channel materials in *V_GS_* = 0.5 V; (**b**) transfer characteristic curve of different channel materials at *V_DS_* = 2.5 V.

**Figure 5 sensors-20-06921-f005:**
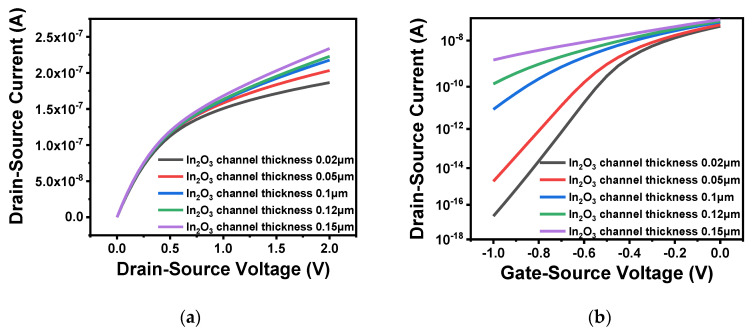
Characteristics of different channel thickness: (**a**) output characteristic curve of different channel thickness; (**b**) transfer characteristic curve of different channel thickness.

**Figure 6 sensors-20-06921-f006:**
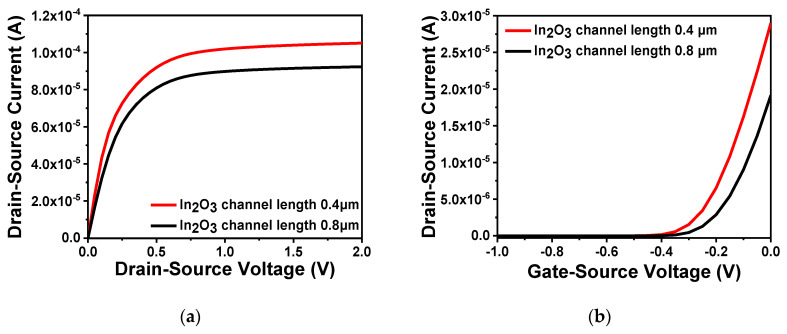
Characteristics of different channel lengths: (**a**) output characteristic curve of different channel lengths; (**b**) transfer characteristic curve of different channel lengths.

**Figure 7 sensors-20-06921-f007:**
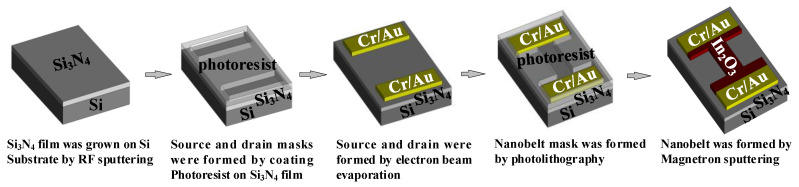
Preparation process of pH-sensitive ion-sensitive field-effect transistor (ISFET).

**Figure 8 sensors-20-06921-f008:**
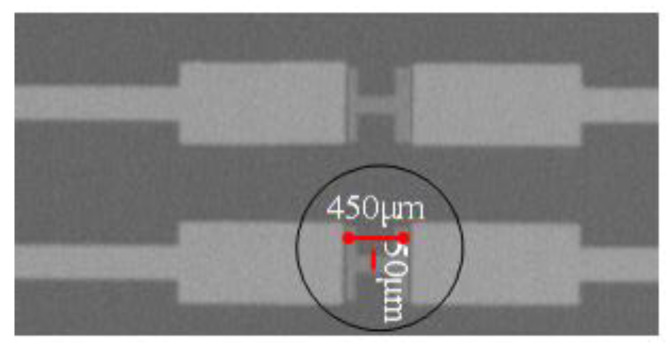
Electron microscopy of ISFET sensing area.

**Figure 9 sensors-20-06921-f009:**
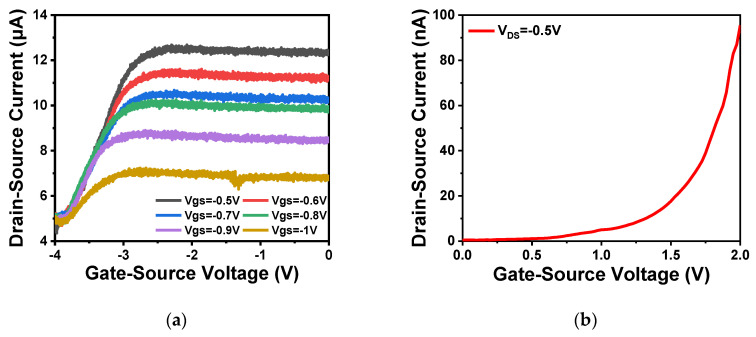
ISFET characteristics of In_2_O_3_ nanobelt: (**a**) ISFET output characteristic curve of In_2_O_3_ nanobelt; (**b**) ISFET transfer characteristic curve of In_2_O_3_ nanobelt.

**Figure 10 sensors-20-06921-f010:**
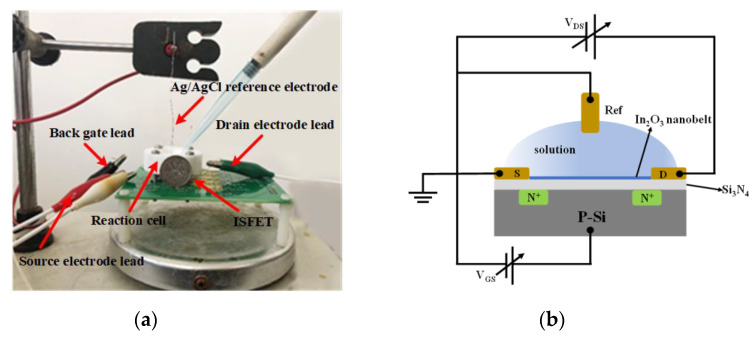
pH detection for micro solution: (**a**) physical map of testing equipment; (**b**) dual-gate (DG) mode electrical connection diagram.

**Figure 11 sensors-20-06921-f011:**
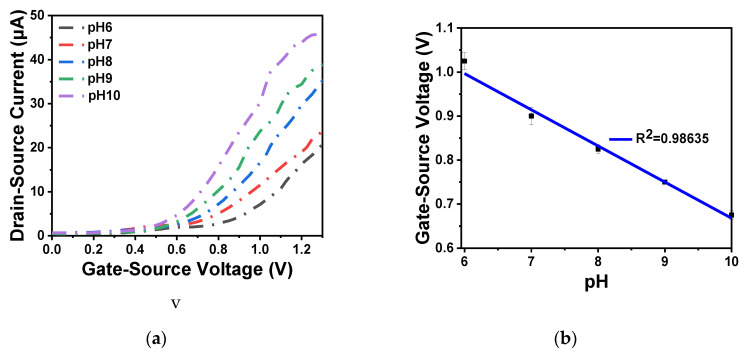
In_2_O_3_ nanobelt ISFET pH detection: (**a**) I_DS_-V_GS_ curve family with pH from 6 to 10; (**b**) linear curve of V_GS_-pH from 6 to 10; (**c**) linear curve of I_DS_-pH from 6 to 10.

**Figure 12 sensors-20-06921-f012:**
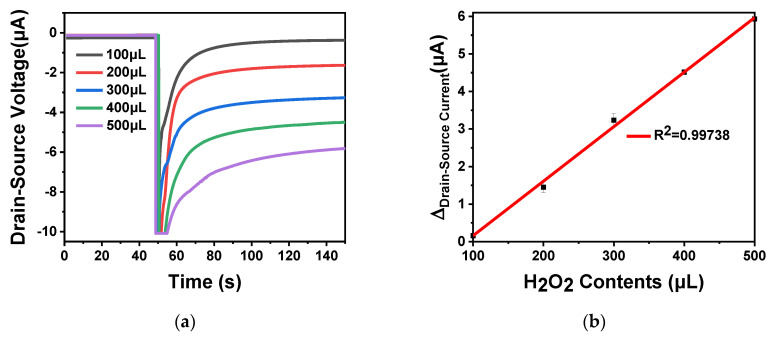
In_2_O_3_ nanobelt ISFET hydrogen peroxide detection: (**a**) the reaction curve of hydrogen peroxide and horseradish peroxidase, respectively add 100, 200, 300, 400 and 500 μL of 0.1 mol L^−1^ hydrogen peroxide and immobilized horseradish peroxidase in 0.01 × PBS environment; (**b**) linear curve of the difference between the drain–source current of the reaction of hydrogen peroxide and horseradish peroxidase.

**Table 1 sensors-20-06921-t001:** Parameters in the simulation.

Parameters	In_2_O_3_	Si_3_N_4_	Si	Gate
permittivity	3.7	7.5	11.8	11.8
thickness	50 nm	50 nm	600 nm	30 nm
length	400 nm	1.2 μm	1.2 μm	1.2 μm

**Table 2 sensors-20-06921-t002:** Subthreshold swing under different thickness.

Thickness (μm)	0.02	0.05	0.1	0.12	0.15
Subthreshold slope (mV/dec)	67.66	81.38	170.14	273.25	535.03

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
