# Peer review of "Design and Implementation of a pH Sensor for Micro Solution Based on Nanostructured Ion-Sensitive Field-Effect Transistor"

_sensors, 2020, doi:10.3390/s20236921_

Round 1

Reviewer 1 Report

Please see detail on the attachment.

Reviewer 2 Report

The manuscript is mainly focus on  two-dimensional simulation of a nanochannel FET was studied using the SILVACO TCAD simulation software and nanostructured ion-sensitive field-effect transistor was developed based Si3N4 was selected as the sensitive film material, In2O3 was used as channel material. Authors carried out excellent study on this work.

Comments#

  1. In conclusion authors mentioned that Si3N4 was selected as the sensitive film material, In2O3 was used as channel material. However, in main manuscript authors mentioned ‘sensitivity of the In2O3 nanobelt ISFET can reach 88.125 mV/pH. It can be seen that the prepared In2O3 nanobelt ISFET can detect the change of pH’ Could you please confirm the sensitive material. As per the result authors observed Super Nernstian response. Could you please explain the reason for Super Nernstian response?
  2. IN modelling section authors explained about drift effect. Could you explain drift and hysteresis issue of your device
  3. As per authors claim, the proposed structure can be used to detect biomolecules based on pH detection,and has promising applications for water and food quality control, chronic disease treatment and industrial production. However, in this paper you mainly explained the sensitive electrode part. To complete the sensor, miniaturized reference electrode is necessary. How you plan to solve this issue.

Round 2

Reviewer 1 Report

  1. I would strongly recommend the authors to provide literature background of double-gate sensing. The double-gate sensing is equivalent to a built-in amplfier through top-gate sensing and back-gate reading. Since this is the technique used in this work, sufficient background information need to be provided.
  2. The reviwer would appreciate the Figure 10 as a photo of the setup. As shown in the photo, a reference electrode seems to be adopted and therefore please provide further information of it, e.g. Ag/AgCl? In addition, please also provide a circuit schematic after Figure 10 or as Figure 10b to provide further details of the pH sensing configuration. Currently, the sensing related information is insufficient but these information are very valuable to audience.
